# The Impact of Microbiota–Immunity–Hormone Interactions on Autoimmune Diseases and Infection

**DOI:** 10.3390/biomedicines12030616

**Published:** 2024-03-08

**Authors:** Serena Martinelli, Giulia Nannini, Fabio Cianchi, Francesco Coratti, Amedeo Amedei

**Affiliations:** 1Department of Clinical and Experimental Medicine, University of Florence, 50139 Florence, Italy; serena.martinelli@unifi.it (S.M.); giulia.nannini@unifi.it (G.N.); fabio.cianchi@unifi.it (F.C.); corattif@gmail.com (F.C.); 2SOD of Interdisciplinary Internal Medicine, Azienda Ospedaliera Universitaria Careggi (AOUC), 50134 Florence, Italy; 3Network of Immunity in Infection, Malignancy and Autoimmunity (NIIMA), Universal Scientific Education and Research Network (USERN), 50139 Florence, Italy

**Keywords:** autoimmune diseases, infections, microbiota, hormones, immune system

## Abstract

Autoimmune diseases are complex multifactorial disorders, and a mixture of genetic and environmental factors play a role in their onset. In recent years, the microbiota has gained attention as it helps to maintain host health and immune homeostasis and is a relevant player in the interaction between our body and the outside world. Alterations (dysbiosis) in its composition or function have been linked to different pathologies, including autoimmune diseases. Among the different microbiota functions, there is the activation/modulation of immune cells that can protect against infections. However, if dysbiosis occurs, it can compromise the host’s ability to protect against pathogens, contributing to the development and progression of autoimmune diseases. In some cases, infections can trigger autoimmune diseases by several mechanisms, including the alteration of gut permeability and the activation of innate immune cells to produce pro-inflammatory cytokines that recruit autoreactive T and B cells. In this complex scenario, we cannot neglect critical hormones’ roles in regulating immune responses. Different hormones, especially estrogens, have been shown to influence the development and progression of autoimmune diseases by modulating the activity and function of the immune system in different ways. In this review, we summarized the main mechanisms of connection between infections, microbiota, immunity, and hormones in autoimmune diseases’ onset and progression given the influence of some infections and hormone levels on their pathogenesis. In detail, we focused on rheumatoid arthritis, multiple sclerosis, and systemic lupus erythematosus.

## 1. Introduction

The majority of interactions between the immune system and the external environment occur within the gastrointestinal (GI) tract, primarily affecting the community of resident microorganisms known as the intestinal microbiota [1]. These microorganisms present a significant source of antigenic diversity, which the host immune system must carefully manage its responses to. The preservation of tolerance and anti-inflammatory responses requires the engagement of a large range of innate and adaptive immune pathways that work together to control microbiota shaping and reduce systemic inflammation [2].

Additionally, the microbiota plays crucial roles in signaling the correct development, education, and epigenetic capabilities of various immune cells [3,4]. This mutual relationship has evolved over thousands of years. However, the rapid modernization of human communities has led to significant changes in environmental exposures and microbiota composition, leading to an increase in autoimmune diseases [4,5]. Autoimmune diseases require a combination of uncontrolled inflammation and self-antigen-specific T cells. Three essential conditions must be met for T cell-mediated autoimmune disorders to manifest: (a) self-reactive T cells must be present and be activated; (b) these T cells must proliferate; and (d) immune regulation must fail to regulate autoreactive responses. Complementarily, hormones, especially estrogens, not only modulate the reproductive system but also regulate immunity development and function. Innate, adaptive, humoral, and cell-mediated immune responses are impacted by hormones, and dysregulation of these mechanisms can contribute to immune-mediated disorders, including autoimmunity [6,7,8].

The aim of our paper is to decipher the complexities of how the microbiota, hormones, and the immune system interact, aiming to assess their collective impact on the onset of autoimmune diseases.

## 2. Microbiota–Immune System Interactions

As previously mentioned, the microbiota is essential for the proper maturation of the host immune system from the earliest stages of life [9]. The immune system has to develop to defend against pathogens while simultaneously tolerating the beneficial microorganisms that coexist symbiotically with the host [10]. Moreover, the microbiota in the large intestine plays a significant role in preserving mucosal and systemic homeostasis. The interaction between the large intestine microbiota and local immune cells is crucial for directing specific immune responses and, consequently, for performing immunomodulatory functions [11]. Notably, the interactions between GM and the immune system established in the first year of life can exert long-term effects on immune responses [12]. This, in turn, may play a role in determining the host’s susceptibility to infections and immune-related disorders later in life [13,14]. In addition, throughout life, GM affects immune functions, often with systemic outcomes that can be independent of the GM colonization site. The GM influences multiple aspects of innate and adaptive immunity. Activation of recognition receptors (PRRs), such as nucleotide-binding oligomerization domain-like receptors (NODs) and Toll-like receptors (TLRs), through commensal bacteria, enhances enterocyte regeneration and survival [15]. The commensal bacterium *Bacteroides fragilis* (*B. fragilis*) produces polysaccharide A (PSA) that recognizes the TLR2/TLR1 heterodimer, inducing the expression of anti-inflammatory genes through cyclic adenosine monophosphate (cAMP)-response element-binding protein (CREB) [16].

In addition, GM can prevent intestinal inflammation by controlling the differentiation of T regulatory (Treg) cells [17]. Metabolites produced by GM, such as short-chain fatty acids (SCFAs) and trimethylamine N-oxide (TMAO), can influence innate and adaptive immune cells in several ways, while the butyrate, through enhancing histone H3 acetylation, induces monocyte-to-macrophage differentiation [18] and TMAO can drive their polarization [19]. Moreover, these molecules reinforce antimicrobial defenses and induce the differentiation of naïve CD4+ into Treg cells [20].

Myeloid differentiation primary response protein (MyD88) serves as an adaptor for various innate immune receptors that detect microbial signals and mediate signaling pathways activated by IL-1 and IL-18 through their respective receptors [3]. Mice lacking MyD88 show a modified microbial composition [21] and MyD88 plays a crucial role in controlling the epithelial expression of several antimicrobial peptides (AMPs), including RegIIIγ. This regulation limits the presence of surface-associated Gram+ bacteria and constrains the activation of adaptive immunity [22]. Additionally, MyD88 influences T cell differentiation, supports microbiota homeostasis by promoting immunoglobulin A (IgA) stimulation, and regulates the differentiation of Th17 cells by inhibiting the growth of segmented filamentous bacteria (SFB) in mice [23].

Of note, GM can also modulate the T helper 17 (Th17) cells; indeed, *Citrobacter* can promote their pro-inflammatory capabilities [24]. Fung et al., show that commensal bacteria residing in lymphoid tissues (LRC) colonized germ-free and antibiotic-treated mice and influenced the cytokines’ production of dendritic cells. This colonization led to the induction of various members of the IL-10 cytokine family, such as dendritic cell-derived IL-10 and group 3 innate lymphoid cell (ILC3)-derived IL-22. As previously reported, IL-10 played a crucial role in limiting pro-inflammatory Th17 cell responses, and IL-22 production contributed to enhanced LRC colonization under steady-state conditions. Those results highlight the straight crosstalk between the host and commensal bacteria [25].

GF colonized by human GM exhibited decreased levels of CD4+ and CD8+ T cells, limited proliferation of T cells, low number of dendritic cells, and decreased expression of antimicrobial peptides in the intestinal tract. Conversely, when GF mice were colonized with SFB derived from mice, the Th17 cell number was restored to levels comparable to those observed in conventionally reared mice (CONV-R mice). These data suggest that specific mice GM may be essential for achieving complete immune maturation in these animals [26].

In addition, gut colonization by SFB elicits IL-17A production by RORγt+ Th17. SFB flagellins stimulate the production of more cytokines, such as IL17, IL21, and IL22, and drive immune endothelial cells (IECs) to secrete serum amyloid (SAA3). These cytokines lately can promote Th17 cell production [27]. Th17 lymphocytes have functional plasticity in response to inflammatory signals; indeed, the presence of high amounts of IL-12 enables them to differentiate in Th17/Th1, while IL-1 and IL-6 can stimulate a Treg-Th17 trans-differentiation [28,29]. These lymphocytes are more pathogenic compared to cells that did not undergo these shifts and can assume a pathogenic role, especially in chronic inflammatory conditions, where inflammation is frequently started by unidentified agents and the immune system lacks the ability to suppress the response [30,31]. On the other hand, Treg cells have a suppressive role (mainly secreting the anti-inflammatory cytokine IL-10). Indeed, they recognize commensal-derived antigens [32], maintain tolerance to intestinal microbes [33], and are essential for suppressing the aberrant activation of myeloid cells and Th17 cells [34]. *Clostridium* species are able to restore the Treg cells’ colonization in germ-free mice through the SCFAs involvement [20,35,36]. Finally, the *Lactobacillus reuteri*, *Lactobacillus murinus*, *Helicobacter hepaticus*, and *B. fragilis* increase the proportion of IL-10-producing Treg cells in mice [17,37]. In other words, the GM composition plays a relevant role in maintaining the proper balance and regulation of T cell subtypes, which is crucial in determining a person’s health status.

## 3. Link between Autoimmunity and Infectious Diseases

Autoimmune diseases (ADs) are a group of chronic and clinically heterogeneous pathologies that affect approximately 5% of the world’s population [38] with a constant increase in westernized societies [39]. Although the understanding of several autoimmune diseases’ pathogenesis still faces open questions, it is usually considered as a result of a mix of genetic and environmental factors. In eubiosis, the gut microbiota (GM) can protect the body against infections through competitive exclusion by contending with pathogenic microorganisms for resources, such as nutrients and space, and so preventing their colonization and growth. In addition, GM secrete antimicrobial compounds, such as bacteriocins and organic acids, that can inhibit the growth and survival of pathogenic microorganisms [40]. However, alterations in GM composition and/or function, such as dysbiosis, can compromise the host’s ability to protect against infections, contributing to the development of infectious diseases [41].

It has been proposed that GM dysbiosis may promote disease onset through infectious pathogens. For example, GM dysbiosis has been linked to various infections, such as *Clostridium difficile*, *Salmonella*, and *Shigella* infections [42]. The GM’s effects on the systemic immune response are mediated by the circulation of microbiota-derived soluble factors from the gut to the periphery [41]. Indeed, GM produce specific molecules (like dsRNAs and peptidoglicans) that can induce the production of cytokines such as interleukin 1 and 6 (IL-1 and IL-6) through the activation of TLRs, promoting the recruitment and activation of immune cells.

These cytokines, especially IL-6, can influence inflammation and regulate adaptive immunity through the induction of Th17 and B cell differentiation [43,44]. Round et al., showed that the PSA of *B. fragilis* can activate Treg cells directly through TLR2 [45].

Moreover, microbiota and its metabolites can induce epigenetic changes. The SCFAs, for example, can inhibit histone deacetylases and stimulate Treg cell differentiation [46]. Notably, the SCFAs play an important role in maintaining a strong gut barrier and in preserving host homeostasis by enhancing the regeneration of epithelial cells, as well as the production of mucus and antimicrobial peptides, preventing infections [47]. Moreover, the SCFAs induce gene expression for B cell differentiation and provide building blocks and energy for antibody production [48].

Anyway, “molecular mimicry” is the leading mechanism through which infectious or chemical substances can trigger autoimmune responses occurring when similarities between foreign and self-peptides lead to the activation of autoreactive T or B cells in susceptible individuals [49,50].

In 1964, Damian used, for the first time, the term “molecular mimicry” to indicate the existence of antigens expressed by infectious agents that were similar to molecules of human hosts that could help microbes avoid the host’s immune response [51]. Two years later, Zabriskie and Freimer observed the similarity between the membrane of *Streptococcus pyogenes* and mammalian muscle [52]. Since their discovery, several pathogens have been documented to carry structurally similar antigens to self-antigens, which activate B and T cells and lead to a crossreactive response against both self- and non-self-antigens [53,54,55].

Finally, more recently, we demonstrated that *Helicobacter pylori* (*H. pylori*)-infected patients with gastric autoimmunity have gastric CD4+ T cells that recognize both H+, K+-adenosine triphosphatase, and *H. pylori* antigens. In addition, we characterized the submolecular specificity of these T cells, identifying crossreactive epitopes from nine *H. pylori* proteins. These peptides were able to induce T cell proliferation and expression of Th- functions [56].

Another mechanism that triggers autoimmunity is the “bystander activation”: a nonspecific and hyperactive antiviral immune response that can create a localized pro-inflammatory environment together with the release of the damaged tissue of self-antigens that can be presented by antigen-presenting cells (APCs) to trigger T cells into an autoreactive state [57,58].

A third way to trigger autoimmunity is “epitope spreading”, where a viral infection can induce the release of new self-antigens that are presented by APCs and activate T cells [59].

In Figure 1, we summarize these three mechanisms of autoimmunity induction, but the infections can also stimulate the secretion of crossreactive antibodies that recognize both the pathogen and the host’s own tissues, leading to tissue damage and inflammation [60].

## 4. The Role of Specific Microorganisms in Some Autoimmune Diseases

As previously mentioned, autoimmune diseases are thought to arise as the result of acquired environmental risk in a genetically susceptible population. Understanding the interaction of environmental factors and genotype is crucial for the development of targeted preventive strategies. These factors can impact the immune system, leading to the aberrant development of plasma cells, the development of autoreactive T cells, and the abnormal production of pro-inflammatory cytokines. The increasing incidence of autoimmune diseases is thought to be a result of substantial GM alterations, influenced by various factors such as dietary shifts and the widespread use of antibiotics.

Among the environmental risks, viruses are the microbial agents that have received the greatest attention for triggering or exacerbating autoimmune diseases.

In detail, Epstein–Barr virus (EBV) has emerged as the virus with the strongest, most consistent, and most biologically plausible association with autoimmunity [61,62]. EBV is a ubiquitous human virus that infects 95% of humans during their lifetime and, after the acute phase, persists for the individual’s whole life. In the latent phase, EBV is prevented from reactivation through efficient cytotoxic cellular immunity, but it can reactivate (lytic phase) under psychological stress conditions, resulting in weakened cellular immunity. EBV chronic activation is a critical mechanism in the pathogenesis of many diseases including autoimmune disorders.

EBV was found to be associated with several autoimmune diseases such as rheumatoid arthritis (RA), systemic lupus erythematosus (SLE), and multiple sclerosis (MS) [63].

However, as previously reported, in addition to infections, GM, as a relevant modulator of immunity and brain function, has emerged as a likely environmental factor contributing to autoimmune diseases. Several mechanisms have been proposed to explain the link between microbiota and autoimmunity; the first of these is the modulation of gut barrier function through the production of various metabolites and signaling molecules, such as lipopolysaccharides (LPSs), SCFAs, and cytokines [64]. Alterations in the gut barrier function can lead to the translocation of microbial antigens and the activation of autoimmune responses. Furthermore, the microbiota can also influence the production of specific immune cell subsets that produce anti-inflammatory cytokines, such as IL-10 and transforming growth factor beta (TGF-β), resulting in the suppression of autoimmune responses. An imbalance in the microbiota composition can lead to the production of pro-inflammatory cytokines, such as tumor necrosis factor alpha (TNF-α) and IL-6, which can promote autoimmune responses [65,66].

### 4.1. Rheumatoid Arthritis

RA is characterized by inflamed and painful joints, which arise from inflammation and thickening of the synovial membrane, leading to the development of excessive connective tissue (known as pannus) and the erosion of bone, ultimately causing disability. Additionally, RA often comes with systemic complications like vascular disorders, osteoporosis, and various other issues [67]. Globally, RA incidence is approximately 1%, and the prevalence increases with age [68,69]; the disease typically onsets between the ages of 40 and 50, with a prevalence three to five times higher in women than in men. RA diagnosis relies on evaluating the patient’s physical symptoms and manifestations [70,71].

Currently, there is a lack of effective treatment, and patients experience the burden of musculoskeletal defects, leading to a decline in physical function and quality of life. RA can be grouped into two main subtypes, namely seropositive and seronegative, depending on the presence or absence of specific serum antibodies related to RA (rheumatoid factor or anticitrullinated peptide antibodies) [72]. Seronegativity is typically defined by the absence of anticitrullinated protein antibodies (ACPA) and/or IgM rheumatoid factor (RF). However, recently, it has been shown that the presence of ACPA or newly discovered autoantibodies, as well as rediagnosis to other rheumatic diseases, is rendering this group extremely heterogenic, and its place in the classification of musculoskeletal diseases remains to be clarified [73,74,75,76]. On the other hand, rheumatoid factor (RF), the classic autoantibody, can be detected in 70–80% of patients with RA, in particular ACPA. The presence of autoantibodies has enabled the recognition of a somewhat homogenous subgroup of patients with certain genetic and environmental risk factors and also a more severe course of the disease [77].

Among the several factors involved in RA pathogenesis are also genetic elements, which include mainly class II major histocompatibility antigens/human leukocyte antigens (HLA-DR), along with non-HLA genes [78]. Smoking and potentially other environmental and lifestyle-related elements may favor the production of ACPA and contribute to the onset of ACPA seropositive RA [79,80].

Moreover, production of pro-inflammatory cytokines and lymphocyte activation are fundamental in the pathogenesis of the disease [81]. In this scenario, IL-17 has been recognized as an essential mediator of cartilage and bone destruction [81,82]. The number of Th17 cells is increased in the early disease stages and in active RA [83,84].

Although factors promoting Th17 differentiation in RA are not fully understood, periodontal pathogens have been described to be implicated in RA etiology [85,86,87]. In addition to the association of EBV infection in RA patients, several studies reported the presence of highly severe forms of periodontal disease (PD) [86,88,89]. Other studies showed a reduction in RA severity when the accompanying PD was successfully treated [90,91]. *Porphyromonas gingivalis* has been described as the main etiological PD agent, and increased antibody titers against *Porphyromonas gingivalis* have been detected in the serum of patients both at high risk of developing RA and in those with RA [92,93]. Notably, the periodontitis induced by *P. gingivalis* and *P. nigrescens* can affect the progression of experimental arthritis in mice, increasing the severity of the induced arthritis [94].

In addition to infectious agents, commensal bacteria have been implicated in RA pathogenesis [95]. Ivanov et al., showed that the introduction of segmented filamentous bacteria (SFB) in GF mice resulted in an increase in Th17 cells in the intestinal lamina propria, promoting the development of autoimmune diseases such as experimental RA [96,97]. Introduction of *B. fragilis* into GF mice, instead, has been shown to induce the correct development of the immune system and induced Treg cells, preventing the occurrence of colitis, meaning that commensal bacteria can reshape the T cell subset and can drive the immune response [45,98,99]. Moreover, in a clinical study, the authors found a strong correlation between the stool presence of *Prevotella copri* with disease in new-onset untreated RA patients [100].

In order to define a microbial and metabolite profile that could predict disease RA status, Chen et al., found that RA patients showed reduced GM diversity compared to controls that correlated with disease and with the levels of autoantibody. In detail, *Collinsella*, *Eggerthella*, and *Faecalibacterium* genera were segregated with RA, and *Collinsella* strongly correlated with high levels of IL-17, suggesting a potential role in altering gut permeability [101].

Moreover, Wu et al., found a decrease in microbial diversity in RA patients’ stool samples compared with healthy subjects, including a lower *Firmicutes*/*Bacteroidetes* (F/B) ratio [102] and depletion of butyrate-producing taxa (*Faecalibacterium*, *Roseburia*, *Subdoligranulum*, *Ruminococcus*, and *Pseudobutyrivibrio*). Intriguingly, the abundance of *Roseburia* negatively correlated with erythrocyte sedimentation rate (ESR) and with blood levels of rheumatoid factors (IgM) in RA patients [102].

Finally, Wang et al., performed a data-driven analysis of the gut microbiome–immune–joint interactions in RA, documenting that GM metabolites were implicated in RA at genetic, functional, and phenotypic levels [103].

In conclusion, these studies demonstrate that the GM plays a fundamental role in maintaining the balance between pro- and anti-inflammatory T cells, thus preserving intestinal homeostasis and influencing disease progression. Table 1 summarizes the findings on the role of microorganisms in RA. In addition, modifications in the dental, gut, or saliva microbiota can discriminate RA patients from healthy controls, and since these changes were correlated with clinical measures [89], the microbiota signature could be used to stratify RA patients on the basis of their response to therapy.

### 4.2. Multiple Sclerosis

Multiple sclerosis (MS) is the most common autoimmune inflammatory demyelinating disease of the central nervous system (CNS), with onset usually between the ages of 20 and 50, affecting more than 2 million people worldwide [104,105]. It is characterized by motor and sensory disturbances associated with vision and cognitive impairment. Three clinical courses of the disease are described: relapsing–remitting (alternating episodes of neurological disability and recovery), primary progressive (gradual worsening from onset), and secondary progressive (relapsing–remitting at the onset but gradual worsening over the MS course) [106,107]. MS etiology is complex and involves the interaction between known susceptibility genes and environmental factors, including infectious agents, lack of sun and vitamin D exposure, smoking, and obesity [108]. Regarding genetic factors, in a recent genome-wide association study (GWAS), 233 single-nucleotide polymorphisms (SNPs or loci) were found to be linked to susceptibility to MS onset. Among these, 32 loci were located within the major histocompatibility complex (MHC), and one locus was identified on the X chromosome. Other SNPs are located within or in close proximity to genes implicated in both the adaptive and innate systems [109,110].

Among the supposed causative factors, the leading candidate is EBV whose contributing role is supported by the increased MS risk after infectious mononucleosis [111], by increased antibody titers against EBV nuclear antigens (EBNAs) in the serum [112], and by the occurrence of EBV in demyelinated lesions [113,114,115]. Finally, Bjornevik and colleagues recently revealed a 32-fold increase in MS risk following EBV infection, with no corresponding increase observed after infection with other viruses, including the similarly transmitted cytomegalovirus. In addition, serum neurofilament light chain levels, a biomarker linked with neuroaxonal degeneration, showed an increase only after seroconversion to EBV. These findings suggest that EBV is the primary MS cause [116].

Additionally, Lanz et al., demonstrated that molecular mimicry between the EBV transcription factor Epstein–Barr nuclear antigen (EBNA1) and glial cell adhesion molecule (GlialCAM) may be the missing molecular link [117]. Indeed, the sequence analysis of immunoglobulin chains from cerebrospinal fluid B cells isolated from nine MS patients showed extensive clonality, suggesting an antigen-specific proliferation. Notably, the B cell-encoded antibodies recognized viral proteins and peptides, particularly EBNA1, which was linked to MS on an epidemiological basis. These findings provide a mechanistic link between EBV infection and the pathobiology of MS [117,118].

As a modulator of the immune response, GM is at the center of research on MS development. Zhou et al., studied the GM of 576 pairs of MS patients and genetically unrelated healthy controls, and they found no difference in α-diversity between MS patients and healthy individuals but a significant difference in β-diversity in disease status. They also did not observe differences in β-diversity between untreated MS and treated MS, suggesting that disease can exert a stronger effect on GM than treatment. On the other hand, *Faecalibacterium prausnitzii* and other beneficial bacteria that secrete metabolites that block nuclear factor kappa B (NF-κB) and IL-8 activation and upregulate Treg cell differentiation [119] were found to be significantly reduced in untreated MS patients. This depletion had a consequential impact on key metabolic pathways, which could potentially worsen MS-associated inflammation [120]. *Streptococcus thermophilus*, *Azospirillum* sp. *47_25*, and *Rhodospirillum* sp. *UNK.MSG-17* were then associated with disease severity [120]. Vice versa, the *Butyrivibrio*, *Clostridium*, and *Ruminococcus* species, which are SCFA producers, correlated with lower MS severity [120]. Since SCFAs have well-documented anti-inflammatory properties, these data suggest that the above-mentioned bacteria have the potential to confer benefits through the production of anti-inflammatory metabolites.

Cox et al., found that β-diversity was significantly different between MS patients and controls, but these differences were not observed between relapsing–remitting and progressive MS patients [121]. In both progressive and relapsing–remitting forms, they observed an increase in *Clostridium bolteae*, *Ruthenibacterium lactatiformans*, and *Akkermansia*, along with a decrease in *Blautia wexlerae*, *Dorea formicigenerans*, and *Erysipelotrichaceae CCM*. Notably, in progressive MS, there were unique findings of elevated Enterobacteriaceae and *Clostridium g24 FCEY*, along with a decrease in *Blautia* and *Agathobaculum*. Additionally, various *Clostridium* species were identified [121].

In a matched case and control longitudinal study, Cantoni and colleagues [122] observed a lower presence of specific bacteria such as *Faecalibacteria*, *Prevotella*, *Lachnospiraceae*, and *Anaerostipes* in MS patients compared to healthy controls, supporting previous research findings [123]. This observation is biologically plausible because these bacteria are known to produce butyrate, which, by activating G protein-coupled receptors and inhibiting histone deacetylase, plays a crucial role in suppressing the demyelination of the CNS, a prominent feature in MS [124]. Previous studies have also revealed reduced levels of SCFAs, including acetate, butyrate, and propionate, in the feces of relapsing–remitting MS patients compared to those without MS [125,126]. A trend towards lower concentrations of butyrate in the stools of MS patients was observed, aligning with the decreased presence of SCFA-producing bacteria in MS. Remarkably, the dietary choices, such as higher meat consumption among MS patients, may contribute to the observed decline in SCFA levels [122]. Castillo-Álvarez et al., at the phylum level, reported statistically significant changes in the abundance of Firmicutes, Proteobacteria, Actinobacteria, and Lentisphaerae between MS patients and controls [127]. The operational taxonomic units (OTUs) analysis revealed that, among these taxa, seven belonged to the phylum of Bacteroidetes, two to Actinobacteria, one to Proteobacteria, and one to Firmicutes. On the contrary, five OTUs (uncultured *Bacteroides* sp.; *Prevotella copri*; uncultured *alpha Proteobacterium*; *Eubacterium eligens*; and uncultured *Pseudomonas* sp.). were less abundant among MS patients. Among these, two were classified under Bacteroidetes, two under Proteobacteria, and one under Firmicutes [127]. The Firmicutes phylum plays a significant role in generating SCFAs, notably butyrate, and it contributes to the differentiation of Treg cells [128,129]. Significant differences in the decrease in *Bacteroides* and increase in *Methanobrevibacter*, *Streptococcus*, and *Akkermansia* abundances were documented in MS patients compared with healthy controls [130,131]. In mice models, some species of *Bifidobacterium* and *Streptococcus* can induce Th17 cells, while *Streptococcus mitis* can induce Th17 cell differentiation in humans [132,133], suggesting that increasing these two species in MS patients could increase the activity of Th17 cells. Although microbiota-driven Th17 cell activation is a putative trigger of MS, aberrant local inflammatory processes in the brain play also a relevant role in disease progression.

On the other hand, microbiota can induce the activation of Treg cells that maintain immune tolerance by producing SCFAs [35,134], which can stimulate the expression of Foxp3, a transcription factor that is essential for Treg cell development, and inhibit the activation of pro-inflammatory immune cells, such as Th17 cells [134,135].

These findings, summarized in Table 1, hold the promise of paving the way for the development of specific probiotics, designed to rejuvenate the natural balance and functionality of the GM, offering potential benefits for MS patients.

### 4.3. Systemic Lupus Erythematosus

Systemic lupus erythematosus (SLE) is a chronic autoimmune disease affecting multiple systems, characterized by a pattern of relapsing and remitting symptoms.

SLE has a multifactorial origin, involving factors such as genetics, hormones, and environmental exposures [136,137]. Smoking, exposure to silica dust, UV radiation, stress, air pollution, pesticides, and heavy metals are the main environmental risk factors that show some evidence of association with SLE [138,139]. Regarding genetic factors, GWAS have revealed over 100 genetic loci associated with susceptibility to SLE across diverse populations, suggesting that a significant portion of the genetic risk is shared across borders and ethnicities [140,141,142].

It is more prevalent in women of childbearing age, with a female predominance of 9:1. Moreover, women with SLE often show more severe disease manifestations compared to men [143,144]. Characteristic of SLE is the presence of antibodies targeting nuclear and cytoplasmic antigens, along with a range of other autoantibodies. These include anti-Scl-70 antibodies (linked to systemic sclerosis), anti-La and anti-Ro antibodies (detected in Sjogren’s disease), anticardiolipin antibodies, and antiphospholipid antibodies. This antibody profile suggests a comprehensive association between SLE and various other autoimmune diseases.

Dysregulation of innate and adaptive immune cells, other SLE characteristics, can result in excessive activation of T and B cells, increased autoantibodies’ production, and the accumulation of immune complexes in renal tubules, leading to glomerulonephritis and inflammation in several organs [145]. SLE pathophysiology is influenced by a complex interplay of genetic, environmental, hormonal, and other immunoregulatory variables, but the etiology is still not entirely clear [146].

As suggested by recent reports, the GM seems implicated in SLE development and symptom onset. In both SLE animal models and patients, alterations have been identified in various taxa of bacteria, such as *Ruminococcus*, *Lactobacillus*, *Akkermansia*, and *B. fragilis* [147,148]. In detail, Luo et al., found that the GM changed significantly before and after SLE onset in New Zealand Black/White F1 (NZB/W F1) mice [149], while Zhang and colleagues observed a notable reduction in *Lactobacillaceae* abundance and a significant increase in *Lachnospiraceae* in MRL/lpr mice predisposed to SLE [150]. In agreement, another investigation reported a diminished presence of *Lactobacillaceae* in MRL/lpr mice [151], and elevated levels of intestinal *Lactobacillaceae* were linked to the amelioration of SLE symptoms, whereas heightened colonization of *Lachnospiraceae* was correlated with SLE progression [150]. Zegarra-Ruiz et al., reported an increased abundance of *Lactobacillus reuteri* in TLR7.1 Tg mice, and the colonization of *Lactobacillus reuteri* exacerbated systemic autoimmunity in both specific pathogen-free and gnotobiotic conditions [152]. The observed reduced F/B ratio in 6-week-old MRL/lpr mice could potentially contribute to the early disease onset [153]. In addition, Valiente et al., found that NZM2410 mice, when colonized with segmented filamentous bacteria, exhibited a deterioration in glomerulonephritis, along with the deposition of immune complexes in both glomerular and tubular regions and interstitial inflammation [154]. Consequently, GM dysbiosis in SLE mouse models is marked by a decline in beneficial bacteria and some increased detrimental bacteria, correlating with SLE.

Finally, also, human clinical trials showed differences in the GM composition between SLE patients and healthy controls. Wang et al., conducted a comparison between SLE patients and their healthy family members, accounting for living conditions and dietary factors. They revealed that the GM of SLE patients still exhibited differences compared to that of healthy controls. Several studies carried out in various countries worldwide have documented a reduced F/B ratio in the GM of SLE patients when compared to healthy individuals [155,156,157]. In their meta-analysis, Xiang et al., reported an increased abundance of *Enterobacteriaceae* and *Enterococcaceae*, along with a decreased abundance of *Ruminococcaceae* in the GM of SLE patients [158] (Table 1).

**Table 1 biomedicines-12-00616-t001:** Studies reporting microorganisms’ involvement in autoimmunity and proposed roles. We need remark that there is no single factor responsible for activating autoimmunity, but it seems that infections and imbalance in microorganism composition are parts of the multifactorial processes involved in autoimmune onset, which can be influenced by several variables.

Autoimmune Disease	Pathogen	Role	Reference
Rheumatoid arthritis	Epstein–Barr virus	Disease onset	[63]
Rheumatoid arthritis	*Porphyromonas gingivalis*	Increase disease severity	[92,93]
Rheumatoid arthritis	*Prevotella copri*	Correlation with disease onset	[100]
Rheumatoid arthritis	*Firmicutes*/*Bacteroidetes* ratio and butyrate-producing taxa	Decreased in RA stool samples	[102]
Multiple sclerosis	Epstein–Barr virus	Disease onset	[116,118]
Multiple sclerosis	*Streptococcus thermophilus*, *Azospirillum* sp. *47_25*, and *Rhodospirillum* sp. *UNK.MSG-17*	Increase disease severity	[120]
Multiple sclerosis	*Clostridium g24 FCEY*	Present in progressive MS	[121]
Multiple sclerosis	*Clostridium bolteae, Ruthenibacterium lactatiformans*, and *Akkermansia*	Highly present in relapsing–remitting MS	[121]
Multiple sclerosis	*Bacteroidetes*, *Actinobacteria*, *Proteobacteria*, and *Firmicutes*	Significantly more abundant in MS patients compared to healthy controls	[128]
Systemic lupus erythematosus	Epstein–Barr virus	Disease onset	[63]
Systemic lupus erythematosus	*Lachnospiraceae*	SLE progression	[150]
Systemic lupus erythematosus	*Enterobacteriaceae* and *Enterococcaceae*	Increased in SLE stool samples	[158]
Systemic lupus erythematosus	*Firmicutes*/*Bacteroidetes* ratio	Decreased in SLE stool samples	[156,157]

## 5. Sexual Dimorphism in Immunity Modulation

SLE, RA, and MS have a female-to-male disease susceptibility ratio of 9:1, 3:1, and 2:1, respectively [159,160,161]. Although complex and likely multifactorial, this gender dimorphism is partly attributable to differences in the levels and response to sex steroid hormones in males and females. It is demonstrated that castration of males enhanced disease progression in animal models of SLE [162] and type 1 diabetes (T1D) [163]. Administration of androgens to females led to their protection from autoimmune diseases [162,164], and hormone treatment was used in SLE patients’ therapy [165]. Finally, a recent study revealed a metabolic signature of urinary steroids associated with SLE, characterized by a lower level of total androgens observed in patients and a slightly higher level of total estrogens in SLE patients than controls [166].

Regarding MS, both men and women have lower testosterone levels when compared to healthy controls [167], and some studies analyzing testosterone as a therapeutic agent described its neuroprotective effect. Indeed, after 12 months of testosterone treatment, remarkable improvements in auditory tasks and a reduction in cerebral volume loss were recorded [168,169].

Notably, gestation usually protects against autoimmune diseases [170,171] by developing an immune-tolerant condition in which the maternal immune system adapts to the allogeneic tissues of the fetus. Cytokines produced by the fetoplacental unit can modulate maternal immune responses, promoting a strong Th2 and decreasing Th1/Th17-mediated response to reduce the risks of miscarriage [172,173]. Estrogen/estrogen receptor (E2/ER) signaling plays an active role in the development, differentiation, and functionality of both innate and adaptive immune cells [174,175,176]. Indeed, a direct E2 role in regulating the function and differentiation of immune cells has been confirmed both in the healthy immune system and in several diseases [177].

Regarding the effects of E2 on adaptive immunity, it can influence T cell biology throughout their entire life cycle, from right maturation through to the modulation of effector functions since thymocytes and thymic epithelial cells express ER [178,179,180]. Moreover, in mouse models, it has been shown that E2 can trigger thymic atrophy through apoptosis induction in T cells involving Fas–Fas ligand (FasL) interaction [181,182,183]. E2 can also induce an extrathymic pathway of T cell differentiation in the liver (Figure 2). It is thought that these extrathymically produced T cells are more autoreactive and could thus contribute to the higher incidence of autoimmune disease in women [184]. In a mouse model of ER ablation specifically in T lymphocytes, the authors observed an increased T cell activation, proliferation, survival, and Th subset differentiation, demonstrating the ER relevance in regulating T cell functions and suggesting that ER may be a potential therapeutic target for autoimmune disorders [185].

E2 has been demonstrated to increase the abundance of bone marrow progenitor B cells and enhance the survival of splenic B lymphocytes, promoting the development of autoreactive B cells [186].

In the spleen, E2 can promote the expansion of the transient and marginal B zone and follicular B cell pools, losing the criteria for negative selection, thus allowing the development of autoreactive B lymphocytes [187] (Figure 2).

Finally, in a mouse model of SLE, E2 treatment increased autoreactive B cell survival, and cells were likely to be eliminated in the central tolerance process [188]. The molecular features of these cells suggested that E2 treatment enhanced antiapoptotic *Bcl2* gene expression, as well as that of genes like *Shp2* and *Vcam* that are associated with autoreactive B cell survival [186,188].

In MS, B cells act as antigen-presenting cells and produce antimyelin antibodies and cytokines that contribute to the pathogenesis [189]. Comparing mRNA and protein expression of male and female thymus revealed that autoimmune regulator (Aire) levels were higher in males than in females, in mice and in humans [190,191], and in an MS mouse model, androgen administration protected against autoimmunity through Aire-dependent mechanisms. In castrated male mice, sex differences in Aire expression compared to females were lost. These results support an androgen-driven mechanism that contributes to gender differences in autoimmunity reinforcing a central tolerance barrier, which limits the release of autoimmune T cells into the periphery [191].

Regarding androgens, they exert their biological functions by binding to and activating the androgen receptor (AR) [192]. More studies suggest the involvement of androgens/ARs in immunomodulation, influencing both innate and adaptive immunity. Cumulatively, these hormones demonstrate various immunosuppressive effects, such as diminishing antibodies’ production, lowering the count and activation potential of T cells, and promoting the secretion of anti-inflammatory cytokines by antigen-presenting cells [193,194,195]. These hormones, whose levels are higher in males, seem to have a protective role against the development of various immune-inflammatory diseases [194,196]. However, the relationship between androgens and disease activity is still unclear. In RA patients, Cutolo et al., found higher E2 levels and lower testosterone and progesterone levels compared with healthy controls. Accordingly, Gupta and colleagues detected low levels of testosterone, dehydroepiandrosterone sulfate (DHEAS), and androgen/E2 ratio in serum and synovial fluid of RA patients [197,198].

The supposed mechanism is that increased levels of TNF-α, IL-6, and IL-1, famous inflammatory cytokines in RA synovitis, could substantially stimulate aromatase activity in peripheral tissues, thereby converting androgen to E2 [199,200,201]. However, higher serum levels of testosterone and DHEAS may predict low disease activity, with likely lower levels of some cytokines, such as IL-1, IL-6, and TNF-α, which could promote minimum peripheral conversion of androgens to E2 and therefore higher androgens levels [198]. Recently, Wu et al., revealed a distinct steroid profile in patients with SLE marked by elevated levels of three estrogens and two sterols, coupled with a decrease in nine androgens, one corticoid, and two progestins. Notably, the most substantial alterations were observed in androgens, revealing the presence of disorders in the process of androgen-to-estrogen conversion [166]. Within the central nervous system (CNS), dihydrotestosterone inhibits the release of pro-inflammatory factors, such as TNF-α, IL-1β, IL-6, iNOS, COX-2, NO, and PGE2, induced by LPS in primary microglia cells. This inhibition occurs through the suppression of the TLR4-mediated NF-κB and MAPK p38 signaling pathways, protecting neurons from inflammatory damage caused by the activated microglia [202]. Similarly, in animal models, DHEA reduces the T cell response and exhibits anti-inflammatory effects on microglia and astrocytes, thereby alleviating the severity of experimental autoimmune encephalomyelitis (EAE) and inflammation [203,204].

In addition to the sex hormones themselves, males and females also differ in the number of X or Y chromosomes contained in each cell. Mary Lyon suggested that for the maintenance of an equivalent expression of X-coded genes between males and females, one of the X chromosomes in each female cell should be inactivated [205]. X chromosome inactivation (XCI) ensures that females, like males, have a functional copy of the X chromosome in each cell of the body. Because X inactivation is random, in normal females, the maternally inherited X chromosome is active in some cells and the paternally inherited X chromosome is active in other cells [206]. However, some females undergo nonrandom X chromosome silencing, resulting in 80% or more cells of paternal or maternal origin, a phenomenon known as skewed XCI. Notably, distorted XCI is associated with autoimmune diseases. Significant XCI distortion was also observed in patients with rheumatoid arthritis [207].

## 6. Role of Microbiota in Regulating Sex Hormone Levels

It is now clearly recognized that GM is active and functional, exerting effects locally and over long distances with the ability to modulate metabolic and immunological messengers as well as hormone circulating levels, particularly E2 in women [208,209]. The link between GM and E2 was observed as antibiotic assumption has been shown to reduce E2 levels in women [210]. The E2 hydroxylated and conjugated to their metabolites are secreted into the bile and subsequently into the GI tract, where they can be deconjugated into active E2 accordingly by the activity of the β-glucuronidase enzyme. This enzyme is encoded by several GM genera, including *Bacteroides*, *Bifidobacterium*, *Escherichia*, *Fecalibacterium*, *Lactobacillus*, and *Roseburia* [211,212,213], that are also able to modulate systemic E2 and their metabolites’ (hydroxylated species from estrone or estradiol) concentration. Ervin et al., demonstrated that GM β-glucuronidase can reactivate two different estrogen glucuronides, estrone-3-glucuronide and estradiol-17-glucuronide, to estrone and estradiol, respectively, from their inactive glucuronides [212].

In light of these data, the gut can be a reservoir of estrogenic metabolites with a local and distant action capacity affecting both health and disease condition.

In addition to E2 hormones, there are plant compounds, called phytoestrogens, which show structural and functional similarities to E2 [214]. Phytoestrogens include isoflavones, such as genistein and daidzein, mainly abundant in soya, that are activated after being metabolized by the GM through conversion of the isoflavone daidzein to O-desmethylangolensin (ODMA) and equol. Both of them have estrogenic activity and can cause physiological effects by affecting cell signaling and may trigger also epigenetic effects and intracellular signaling cascades [215,216,217,218].

Finally, there are the endocrine disruptors, defined as “exogenous agents that interfere with the synthesis, secretion, transport, metabolism, binding action, or elimination of natural blood-borne hormones that are present in the body and are responsible for homeostasis, reproduction, and developmental process” [219]. By their binding to ER, they can elicit downstream gene activation and trigger intracellular signaling cascades in more tissues, affecting the host metabolism [220]. There is a bidirectional relationship between GM and endocrine disruptors since GM can metabolize the compounds into biologically active or inactive forms; meanwhile, endocrine disruptors can selectively induce the growth of specific GM populations. In detail, *Clostridium methoxybenzovorans* and *Bifidobacterium pseudocatenulatum WC 401* can deglucosylate, respectively, anhydrosecoisolariciresinol and secoisolariciresinol diglucoside [221,222], transforming them into enterodiol and enterolactone, which exert E2 activity. Anhydrosecoisolariciresinol and secoisolariciresinol can be also demethylated by *Peptostreptococcus*, *Eubacterium limosum*, and *Clostridium methoxybenzovorans* [223].

Likewise, it has recently shown that the gut microbiome is implicated in the metabolism and deglucuronidation of dihydrotestosterone (DHT) and testosterone, resulting in exceptionally high DHT levels [224]. Furthermore, a potential GM mechanism to modulate the sex hormones could be the hydroxysteroid dehydrogenase (HSD) enzymes, which are involved in the metabolism of steroid hormones and control steroids binding to their nuclear receptors, causing them to act as activators or inhibitors [225,226].

Finally, the GM can influence the sex hormones’ concentration through SCFA production [156]. SCFAs function by binding to G protein-coupled receptors (free fatty acid receptors (FFARs) 2 and 3) and, through adenylate cyclase, can lead to inhibition of cAMP pathways. G protein activation leads to hydroxylation of phosphatidylinositol 4,5 bisphosphate (PIP2) to 1,2 diacylglycerol (DAG) and inositol 1,4,5 triphosphate (IP3), activating protein kinase C (PKC) and increasing calcium release [227,228].

Overall, the microbiome–hormone interactions play a critical role in modulating the immune system’s activity and function, and alterations in hormones’ levels or signaling can contribute to the development and progression of autoimmune diseases. Further research is needed to elucidate the specific mechanisms involved and to develop novel therapeutic strategies based on hormone modulation by manipulating the microbiota, including fecal microbiota transplantation (FMT) [229].

## 7. Clinical Trials Examining the Role of FMT in Autoimmune Diseases

FMT consists of the transferring of the entire community of human fecal microbiota from a healthy donor to the GI tract of a recipient patient with the aim of re-establishing microbial diversity and host intestinal health [230]. FMT is currently a consolidated treatment for recurrent *Clostridium difficile* infection (CDI) that is not responding to standard therapies, and since 2020, the research has expanded to explore FMT’s potential in a plethora of other pathologies such as neurodegenerative diseases [231] and autoimmune diseases [232].

A study transplanting human fecal material into a mouse MS model showed that mice colonized with microbiota derived from MS patients had a higher frequency of spontaneous experimental autoimmune encephalomyelitis (EAE) than mice transplanted with GM from healthy twins [233]. This study suggested that FMT can be an innovative therapy by modulating the immune response in MS. There are three active clinical trials, one completed and two terminated, on MS, as summarized in Table 2.

Regarding RA, one case report described that a patient with refractory RA was successfully treated with FMT, suggesting its good therapeutic effects on RA [234]. Meanwhile, a clinical trial is reported on the evaluation of FMT efficacy and safety in patients with RA refractory to methotrexate (Table 2).

Regarding SLE treatment, recent studies on mice models documented that the GM derived from SLE patients and transplanted into recipient mice induced the production of autoantibodies and upregulated the expression of genes associated with SLE onset. In addition, Choi et al., transplanting the dysbiotic GM from triple congenic lupus-prone mice into germ-free congenic C57BL/6 mice [235], observed that the transplanted GM activated immune cells and triggered the autoantibodies production in the recipient mice. Similarly, Ma et al., after FMT from SLE mice into germ-free mice, observed that the fecal microbiome from SLE mice stimulated the secretion of anti-dsDNA antibodies and increased the expression of susceptibility genes associated with SLE in germ-free mice [236]. Furthermore, germ-free or germ-depleted mice exhibited elevated blood pressure and vascular complications following the transplantation of GM obtained from hypertensive NZBWF1 mice [237]. Finally, the first clinical trial using FMT for the treatment of SLE patients was conducted by Huang et al., using an oral encapsulated microbiome isolated from the feces of healthy donors [238]. The authors reported that FMT treatment led to a significant reduction in the Systemic Lupus Erythematosus Disease Activity Index 2000 (SLEDAI-2K) score and the levels of serum anti-dsDNA antibodies. Additionally, they observed a significant decrease in bacterial taxa linked to inflammation and an increase in bacteria producing SCFAs. Finally, the peripheral blood levels of IL-6 and the CD4^+^ memory/naïve ratio decreased after FMT, whereas the synthesis of SCFAs increased [238].

## 8. Other GM-Modulating Approaches and Future Perspectives

Regarding GM modulation, in addition to FMT, there are other promising approaches involving the administration of probiotics, prebiotics, symbiotics, and postbiotics.

These natural compounds are nontargeted approaches in GM shaping; however, their use in combination with other therapeutic interventions should be taken into account [239,240]. A synthetic bacterial preparation of microorganisms called “Bacterial Consortium” is under development with the aim to provide the administration of specific beneficial bacterial strains to support the growth of a new community, with the goal of achieving beneficial outcomes [241,242].

Some probiotic strains, including those of *Lactobacilli*, *Bifidobacteria*, *Propionibacterium*, *E. coli*, *Saccharomyces*, and *Bacillus*, can positively regulate TLR activation through the decrease in MAPK activation and NF-κB pathways, thus limiting the production of pro-inflammatory cytokines [243]. Moreover, several small molecules have demonstrated efficacy in inhibiting the bacterial β-glucuronidase enzyme, a pivotal player in metabolizing glucuronide drug conjugates produced by host metabolism [244,245]. This intervention strategy could help in regulating the levels of GM producing E2 in autoimmune diseases, contributing to the limitation of E2 circulating levels. The diet regimes, as well as the administration of biotics, drugs limiting the β-glucuronidase enzyme, and “Bacterial Consortium”, could effectively influence the GM composition. This could enhance the presence of beneficial microbes, preventing infections and restoring a functional gut microbiome, thus contributing to improving patients’ responses to therapies.

## 9. Conclusions

There are biological differences in immunological responses to stimuli and to hormone circulating levels between males and females, and this can contribute to sex differences in the loss of immunological tolerance and autoantibody production. Although estrogens generally protect women from infections, they predispose the same to chronic inflammatory conditions and are a major risk in the development of autoimmunity compared to their male counterparts.

In addition, microbial metabolism may exert protection or promote exacerbation of some disease processes by regulating both sex hormone circulating levels and immune system homeostasis. Given the complexity of the several factors implicated in autoimmune diseases’ onset, a multifaceted approach is needed to treat these pathologies. By employing approaches such as FMT and other treatments to modulate the microbiota, along with methods to regulate hormones, it becomes imperative to advance personalized medicine. We are confident that this progression is crucial for attaining improved therapeutic results in the treatment of autoimmune diseases.

## Figures and Tables

**Figure 1 biomedicines-12-00616-f001:**
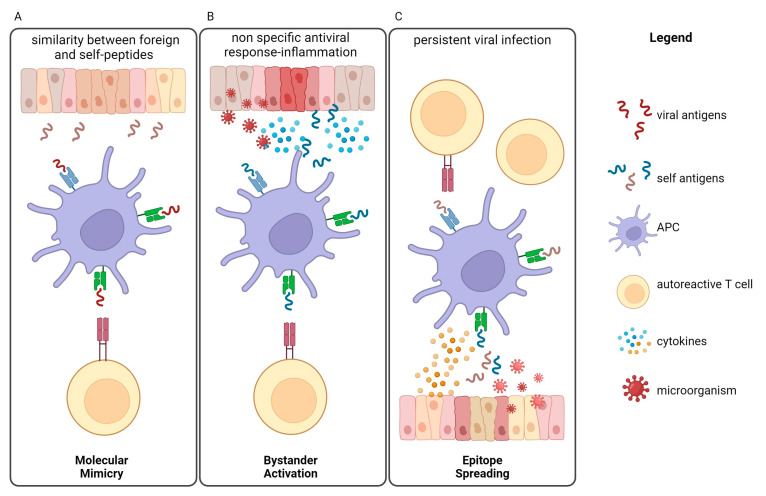
Infections can trigger or exacerbate autoimmune diseases through several mechanisms, leading to autoimmunity induction. (**A**) Molecular mimicry is the mechanism by which infectious antigens similar to self-molecules and presented by APCs can trigger T autoreactive cells, leading to the development of autoimmune diseases. (**B**) Bystander activation refers to the way in which over-reactive antiviral immune responses lead to the release of self-antigens and inflammatory cytokines from damaged tissue. Autoreactive T cells are then activated by APCs. (**C**) The epitope spreading model predicts that a persistent infection induces tissue damage and release of new self-antigens that are presented by APCs. Nonspecific triggering of several autoreactive T cells can lead to autoimmunity. APC = antigen-presenting cell.

**Figure 2 biomedicines-12-00616-f002:**
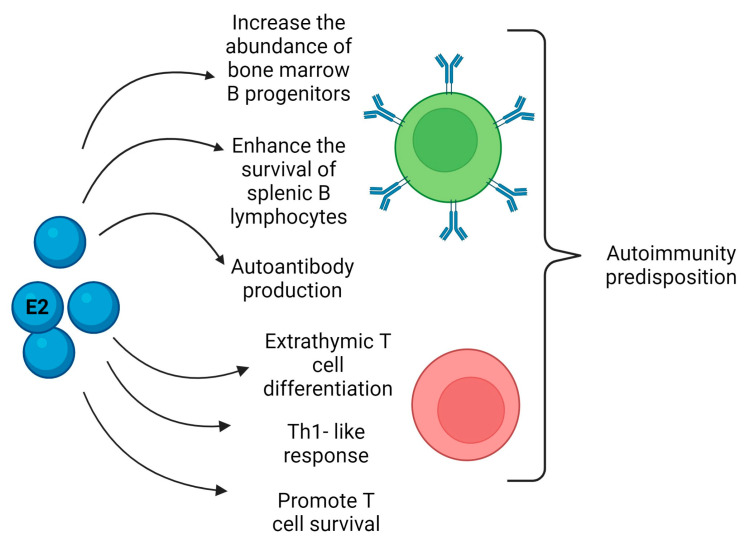
Schematization of estrogen (E2) actions on B (green) and T (red) cells. There are several actions of estrogens on B cells such as the increase in cell number progenitors in the bone marrow, the enhanced survival in the spleen, and the induction of antibody production. Regarding E2’s effects on T cells, the promotion of cell activation, proliferation, survival, and differentiation have been described in Th-1 subtype. Extrathymic cell differentiation in the liver was observed. All these features could lead to the predisposition to autoimmunity and disease development when an imbalance occurs.

**Table 2 biomedicines-12-00616-t002:** Clinical trials involving FMT in multiple sclerosis, rheumatoid arthritis, and systemic lupus erythematosus.

Disease	Study Title	Clinical Trial ID	Status of the Study	Study Start
Multiple Sclerosis	Fecal Microbial Transplantation in Relapsing Multiple Sclerosis Patients	NCT03183869	Terminated with results	24 August 2017
Multiple Sclerosis	Fecal Microbiota Transplantation After Autologous HSCT in Patients with Multiple Sclerosis	NCT04203017	Terminated because of corrupted biological samples	6 December 2023
Multiple Sclerosis	Fecal Microbiota Transplantation (FMT) in Multiple Sclerosis	NCT03975413	Completed	8 October 2020
Multiple Sclerosis	Safety and Efficacy of Fecal Microbiota Transplantation	NCT04014413	Active, recruiting	30 May 2023
Multiple Sclerosis	Fecal Microbiota Transplantation (FMT) of FMP30 in Relapsing–Remitting Multiple Sclerosis (MS-BIOME)	NCT03594487	Active, not recruiting	3 July 2023
Rheumatoid Arthritis	Efficacy and Safety of Fecal Microbiota Transplantation in Patients With Rheumatoid Arthritis Refractory to Methotrexate (FARM)	NCT03944096	Unknown status	30 April 2019
Systemic Lupus Erythematosus	Safety and Efficacy of Fecal Microbiota Transplantation for Treatment of Systemic Lupus Erythematosus: An EXPLORER Trial	ChiCTR2000036352	Completed	22 August 2020

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
