# Peer review of "The Impact of Microbiota–Immunity–Hormone Interactions on Autoimmune Diseases and Infection"

_biomedicines, 2024, doi:10.3390/biomedicines12030616_

Round 1

Reviewer 1 Report

Comments and Suggestions for Authors

The review titled “Infection and autoimmunity: the impact of hormone-microbiota-immunity threesome” by Serena Martinelli al., aims to summarize the presence information for the main mechanisms of connection between gut microbiota, immunity, and sex hormones.

The manuscript is well-written but there are some comments and remarks to the authors:

There are spelling mistakes in the text – lines 73, 74, 151, 234, 242, 530

Figure 1 B and C are almost the same.

The paragraph from line 355 to line 366 could be transposed to and integrated with section 6.

Figure 2 – spelling mistake - vell instead of cell

Author Response

We thank the reviewers for their right comments on the manuscript and the positive evaluation of our work. We have reviewed and modified the manuscript improved it according to the criticisms of the reviewers.

Please find attached a revised version of our manuscript entitled “Autoimmune diseases: the impact of infections and hormone-microbiota-immunity threesome” with the revisions highlighted in yellow

We hope that the revisions in our manuscript and our accompanying replies will be adequate to make our manuscript suitable for publication.

Reviewer 2 Report

Comments and Suggestions for Authors

The manuscript by Martinelli and colleagues aims to summarize the knowledge that links infections, sex hormones and gut microbiota with autoimmune diseases. The article is interesting and well structured. However, there are several points to consider before publication.

My main remarks envision the title and the abstract. They are misleading because they highlight both infectious and autoimmune diseases while the main text is concentrated on autoimmunity and the impact of hormones microbiota and infections on autoimmunity onset and progression. A more appropriate title for this article will be: "Autoimmune diseases: the impact of hormone-microbiota-infections threesome". The abstract should be revised in order to better describe the aim of this review and its main focus.

Minor points:

1) L10-11: Split the sentence. "The microbiota plays a critical role in maintaining host health and immune homeostasis. Alterations in its composition (dysbiosis) or function have been linked to different pathologies..."

2) L16-17: replace "anti-pathogens' protection" with a more appropriate term.

3) It should be mentioned in the abstract that three main autimmune diseases are discussed in regard to the influence of some infections on their pathogenesis.

4) L37-38: delete "which includes bacteria, fungi, helminths, and viruses.".

5) L48: delete "and infectious". Infectious diseases do not require self-antigen-specific T cells.

6) L73-74: revise the sentence.

7) L83: write the full name of B. fragilis when metioned for the first time in the text. The same remark is valid for L105 - H. pylori.

8) Check all latin names of different species and genera and italicize them (L107, L108, L281, L307, L344-346, L464, L515 etc.)

9) Revise figure 1. The scheme for molecular mimicry is incomplete. The text in the upper part of schemes B and C is too small.

10) Include a table that summarizes the findings for role of specific microorganisms in some autoimmune diseases.

11) L173-174: revise the sentence

12) L227: Replace "ongoing" with progression. L234: Should be 'It is' instead of "Is". L264: 'exert' instead of "exerted". L283: delete "by". 

13) L324: Incomplete sentence. Revise. L370: 'revealed' instead of "evealed". L428: It should be 'in mice'.

14) L443: XCI - define the abbreviation. L465: 'also' instead of "so". L526: delete "the".

Comments on the Quality of English Language

The comments are included in the upper section.

Author Response

(The authors gave the same response as above.)

Reviewer 3 Report

Comments and Suggestions for Authors

Infection and Autoimmunity: The Impact of Hormone-Microbiota-Immunity Threesome" would focus on its scientific merit and contribution to the field. The article presents a comprehensive review of the intricate interplay between microbiota, hormonal regulation, and immune responses, particularly highlighting the role of dysbiosis in autoimmune diseases. The connection between the gut microbiota and the immune system, especially the differentiation of Tregs and the modulation of gut barrier function, is well-articulated. Moreover, the article insightfully discusses how hormonal influences, particularly estrogens, affect autoimmune disease progression. However, the review could benefit from a more detailed exploration of the molecular mechanisms underlying these interactions. Furthermore, while the article provides a broad overview, it lacks specific examples of diseases where this tripartite interaction plays a crucial role.

Some questions arising from this review are:

1. Why is there a specific focus on estrogens in the hormonal regulation of autoimmune diseases?

2. Why does the article not delve deeper into the molecular mechanisms that govern the microbiota-immune system interactions?

3. Why are specific autoimmune diseases not exemplified to illustrate the practical implications of these interactions?

Comments on the Quality of English Language

The article is generally well-written, exhibiting a high standard of academic English. The vocabulary is appropriate for a scientific audience, with technical terms used accurately and effectively.

Author Response

(The authors gave the same response as above.)

Round 2

Reviewer 3 Report

Comments and Suggestions for Authors

I am pleased to acknowledge the efforts made to address the concerns raised in the evaluation of the manuscript titled "Infection and Autoimmunity: The Impact of Hormone-Microbiota-Immunity Threesome." The authors have made commendable strides in enhancing the scientific merit and contribution of their article to the field of immunology and microbiota research. Herein, I provide a further evaluation based on the authors' revisions:

Point 1 Response Evaluation:

The authors' clarification on the emphasis on estrogens over androgens in the context of hormonal regulation and its impact on autoimmune diseases is well-received. Their rationale, that estrogens have a more pronounced link with immune modulation and are thus more relevant to the pathogenesis of autoimmune diseases, is scientifically sound. The mention of estrogen concentration regulation by specific commensal bacterial strains adds an important dimension to the discourse on hormone-microbiota-immunity interplay. However, it would be beneficial for the article to briefly discuss the roles of other hormones, even if they are not as central as estrogens, to provide a holistic view of hormonal influences on autoimmunity.

Point 2 Response Evaluation:

The addition of a paragraph detailing the molecular mechanisms through which the microbiota modulates the immune system significantly strengthens the article. This expansion (lines 60-107) helps in bridging the gap previously noted regarding the depth of discussion on microbiota-immune system interactions. It is commendable that the authors took this feedback into account and enriched the manuscript with crucial mechanistic insights. Nonetheless, for further enhancement, it would be advantageous if the authors could incorporate recent studies or examples that illustrate these mechanisms in vivo or in clinical settings.

Point 3 Response Evaluation:

The incorporation of a new section titled "Microbiota-immune system interactions," specifically aimed at clarifying the implications of these interactions in the development of autoimmune diseases, addresses the previous lack of specific disease examples. This addition is a significant improvement, as it directly ties the theoretical framework presented in the article to practical, disease-related outcomes. To further augment this section, the authors might consider including case studies or clinical trial results that demonstrate the therapeutic potential of modulating the microbiota-immune-hormone axis in treating or managing autoimmune diseases.

Further Suggestions:

While the authors have addressed the primary concerns, I recommend the following for further refinement:

Comparative Analysis: A brief comparison with other regulatory mechanisms or factors influencing autoimmune disease progression could provide readers with a comprehensive understanding of the subject matter's complexity.

Future Directions: Expanding on potential therapeutic interventions or future research directions based on the hormone-microbiota-immunity interaction could offer valuable insights for advancing the field.

In conclusion, the revisions made by the authors have substantially improved the article's quality and its scientific contribution. The manuscript now presents a more detailed and nuanced exploration of the hormone-microbiota-immunity threesome's role in autoimmune diseases. With the suggested enhancements, this work could serve as a significant reference for researchers and clinicians interested in the intersections of microbiology, immunology, and endocrinology.

Comments on the Quality of English Language

The quality of English used throughout the document is of a high standard

Author Response

Dear Editor,

We thank the Reviewer for the careful reading and critical comments that significantly improved the manuscript. Changes in the text has been highlighted throughout the manuscript.

Reviewer 3:

I am pleased to acknowledge the efforts made to address the concerns raised in the evaluation of the manuscript titled "Infection and Autoimmunity: The Impact of Hormone-Microbiota-Immunity Threesome." The authors have made commendable strides in enhancing the scientific merit and contribution of their article to the field of immunology and microbiota research. Herein, I provide a further evaluation based on the authors' revisions:

Point 1 Response Evaluation:

The authors' clarification on the emphasis on estrogens over androgens in the context of hormonal regulation and its impact on autoimmune diseases is well-received. Their rationale, that estrogens have a more pronounced link with immune modulation and are thus more relevant to the pathogenesis of autoimmune diseases, is scientifically sound. The mention of estrogen concentration regulation by specific commensal bacterial strains adds an important dimension to the discourse on hormone-microbiota-immunity interplay. However, it would be beneficial for the article to briefly discuss the roles of other hormones, even if they are not as central as estrogens, to provide a holistic view of hormonal influences on autoimmunity.

Reply1:  Following the reviewer suggestions, we discussed in section 5 on the impact of androgen hormones in autoimmunity (please see the lines 534- 565).

Point 2 Response Evaluation:

The addition of a paragraph detailing the molecular mechanisms through which the microbiota modulates the immune system significantly strengthens the article. This expansion (lines 60-107) helps in bridging the gap previously noted regarding the depth of discussion on microbiota-immune system interactions. It is commendable that the authors took this feedback into account and enriched the manuscript with crucial mechanistic insights. Nonetheless, for further enhancement, it would be advantageous if the authors could incorporate recent studies or examples that illustrate these mechanisms in vivo or in clinical settings.

Reply2: As rightly suggested by the reviewer we included animal models’ studies to better explain the microbiota-immune interactions (please see the lines 87-96 and 98-112).

Point 3 Response Evaluation:

The incorporation of a new section titled "Microbiota-immune system interactions," specifically aimed at clarifying the implications of these interactions in the development of autoimmune diseases, addresses the previous lack of specific disease examples. This addition is a significant improvement, as it directly ties the theoretical framework presented in the article to practical, disease-related outcomes. To further augment this section, the authors might consider including case studies or clinical trial results that demonstrate the therapeutic potential of modulating the microbiota-immune-hormone axis in treating or managing autoimmune diseases.

Reply3: We thank the reviewer for the suggestions. We added some in vitro studies to better explain the GM role on immunity modulation. Regarding the clinical studies, we finely discussed the GM manipulating in clinical trial focused on autoimmune disorders (please see the section 7 and especially the Table 2). Finally, we added a new paragraph (please see the lines 687- 707) to summarize the different strategies of microbiota modulating.

Further Suggestions:

While the authors have addressed the primary concerns, I recommend the following for further refinement:

Point4: Comparative Analysis: A brief comparison with other regulatory mechanisms or factors influencing autoimmune disease progression could provide readers with a comprehensive understanding of the subject matter's complexity.

Reply4: Thanking the review for the critical comment, we additionally discussed genetic and environmental factors involved in the onset of the autoimmune pathologies (please see the lines 212-217; 271-275; 333-339 and 422-428).

Point5: Future Directions: Expanding on potential therapeutic interventions or future research directions based on the hormone-microbiota-immunity interaction could offer valuable insights for advancing the field.

Reply5: As previously reported (see the Reply 3), we included the paragraph 8 describing the different microbiota modulating approaches that could contribute to of the future treatments of patients affected by autoimmune diseases (please see the lines 687- 707).

In conclusion, the revisions made by the authors have substantially improved the article's quality and its scientific contribution. The manuscript now presents a more detailed and nuanced exploration of the hormone-microbiota-immunity threesome's role in autoimmune diseases. With the suggested enhancements, this work could serve as a significant reference for researchers and clinicians interested in the intersections of microbiology, immunology, and endocrinology.
